# Effect of Growth Stages and Lactic Acid Fermentation on Anti-Nutrients and Nutritional Attributes of Spinach (*Spinacia oleracea*)

**DOI:** 10.3390/microorganisms11092343

**Published:** 2023-09-19

**Authors:** Adila Naseem, Saeed Akhtar, Tariq Ismail, Muhammad Qamar, Dur-e-shahwar Sattar, Wisha Saeed, Tuba Esatbeyoglu, Elena Bartkiene, João Miguel Rocha

**Affiliations:** 1Department of Food Science and Nutrition, Faculty of Food Science and Nutrition, Bahauddin Zakariya University, Multan 60000, Pakistan; adilanaseem10@gmail.com (A.N.); saeedakhtar@bzu.edu.pk (S.A.); muhammad.qamar44@gmail.com (M.Q.); dsattar@bzu.edu.pk (D.-e.-s.S.); wishasaeed1@gmail.com (W.S.); 2Department of Food Development and Food Quality, Institute of Food Science and Human Nutrition, Gottfried Wilhelm Leibniz University Hannover, Am Kleinen Felde 30, 30167 Hannover, Germany; 3Department of Food Safety and Quality, Faculty of Veterinary, Lithuanian University of Health Sciences, Tilzes Str. 18, LT-47181 Kaunas, Lithuania; elena.bartkiene@lsmu.lt; 4Faculty of Animal Sciences, Institute of Animal Rearing Technologies, Lithuanian University of Health Sciences, Tilzes Str. 18, LT-47181 Kaunas, Lithuania; 5Universidade Católica Portuguesa, CBQF—Centro de Biotecnologia e Química Fina—Laboratório Associado, Escola Superior de Biotecnologia, Rua Diogo Botelho 1327, 4169-005 Porto, Portugal; jmfrocha@fc.up.pt; 6LEPABE—Laboratory for Process Engineering, Environment, Biotechnology and Energy, Faculty of Engineering, University of Porto, Rua Dr. Roberto Frias, s/n, 4200-465 Porto, Portugal; 7ALiCE—Associate Laboratory in Chemical Engineering, Faculty of Engineering, University of Porto, Rua Dr. Roberto Frias, s/n, 4200-465 Porto, Portugal

**Keywords:** *Spinacia oleracea*, growth stages, fermentation, antinutrient, antioxidant activity, glucosinolate

## Abstract

Spinach (*Spinacia oleracea*) is a winter-season green, leafy vegetable grown all over the world, belonging to the family *Amaranthus*, sub-family *Chenopodiaceae*. Spinach is a low-caloric food and an enormous source of micronutrients, e.g., calcium, folates, zinc, retinol, iron, ascorbic acid and magnesium. Contrarily, it also contains a variety of anti-nutritional factors, e.g., alkaloids, phytates, saponins, oxalates, tannins and many other natural toxicants which may hinder nutrient-absorption. This study was aimed at investigating the effect of fermentation on improving the nutrient-delivering potential of spinach and mitigating its burden of antinutrients and toxicants at three growth stages: the 1st growth stage as baby leaves, the 2nd growth stage at the coarse stage, and the 3rd growth stage at maturation. The results revealed the significant (*p* < 0.05) effect of fermentation on increasing the protein and fiber content of spinach powder from 2.53 to 3.53% and 19.33 to 22.03%, respectively, and on reducing total carbohydrate content from 52.92 to 40.52%; the effect was consistent in all three growth stages. A significant decline in alkaloids (6.45 to 2.20 mg/100 g), oxalates (0.07 mg/100 g to 0.02 mg/100 g), phytates (1.97 to 0.43 mg/100 g) and glucosinolates (201 to 10.50 µmol/g) was observed as a result of fermentation using *Lactiplantibacillus plantarum*. Fermentation had no impact on total phenolic content and the antioxidant potential of spinach, as evaluated using 2,2-diphenyl-1-picrylhydrazyl (DPPH) and ferric-reducing antioxidant power (FRAP) assays. This study proposes fermentation as a safer bioprocess for improving the nutrient-delivering potential of spinach, and suggests processed powders made from spinach as a cost-effective complement to existing plant proteins.

## 1. Introduction

Spinach (*Spinacia oleracea*) is a winter-season green, leafy vegetable grown all over the world, belonging to the family *Amaranthus* sub-family *Chenopodiaceae*. In 2018, the global production of spinach was 26.3 million tons [1]. It is a famous and inexpensive leafy vegetable that is eaten fresh, boiled, cooked and as an ingredient of choice for the development of value-added baked products. Spinach is a low-caloric food and an enormous source of micronutrients (e.g., calcium, folates, zinc, retinol, iron, ascorbic acid and magnesium) for consumers with marginal nutritional status [2]. The World Health Organization (WHO) technical consulting panel and nutritionists from all over the world recommend the dietary diversity that comes from including yellow and dark-green vegetables in a regular diet as the most effective approach to increasing consumption of vitamin A, folates and iron [3].

Leafy green vegetables are abundant in micro- and macronutrients like phytochemicals, vitamins, minerals, fat, protein and numerous other bioactive metabolites [4]. Contrarily, leafy vegetables are also comprised of a variety of anti-nutritional factors, e.g., alkaloids, phytates, saponins, oxalates, tannins and many other natural toxicants, which may hinder nutrient-absorption [5]. Moreover, a dietary burden of extrinsic contaminants is reflective of poor agricultural practices and phytosanitary standards, ignoring goods and environmental toxicants that also affect the nutritional quality of consumable food items [6].

Fermented foods and beverages have been considered to improve nutrition, extend shelf-life and reduce anti-nutritional factors. In recent times, lactic acid fermentation using lactic acid bacteria (LAB) has been extensively adopted industrially for the processing of vegetables and fruit. Fermentation preserves the nutritional value of vegetables without negatively affecting their physical structure or chemical composition [7].

Previous investigations have revealed the greater commercialization potential of non-dairy-based fermented food products due to their better health benefits and high nutritional value [8]. Fermented foods not only contain probiotics, but they also have powerful antioxidant, antibacterial, anti-cancer, antidiabetic, and anti-inflammatory properties [9,10]. Fermented foods are an adequate source of vitamins, total phenols and amino acids with unique sensory characteristics. All these functionalities are associated with plant matrices, and fermentation starter bacterial strains like LAB species [11]. However, the existence of inherent toxic elements or antinutritional elements in vegetables has been a major impediment to reaping the full nutritional benefits of plant food, including vegetables. Although these antinutritional elements are constantly present in small quantities, they have been revealed to perform a major role in the nutritional quality of food [11].

In recent years, fermentation technologies based on the inoculation of various probiotic bacteria like *Lactiplantibacillus plantarum (Lp. Plantarum)* have played an important role in lowering antinutritional factors and increasing the nutritional profiles of fermented goods [12,13,14,15,16]. However, data on the application of microbial strain (starter cultures) like *Lp. Plantarum* during fermentation of spinach are scanty [17]. Importantly, different growth stages of spinach are reported to show a difference in nutritional composition i.e., the most suitable time for utilizing spinach at maturity delivered the highest amount of nutrients as compared to the baby leaves [18] This current study aims to use a cost-effective microbial fermentation approach to minimize the burden of intrinsic toxicants in spinach at different growth stages, and to improve nutritional bioavailability and increase nutrient-delivering properties. In the present investigation, the *Lp. plantarum* strain was employed and studied for its effect on antioxidant content, and the antinutritional and proximate composition of fermented or raw spinach powder.

## 2. Materials and Methods

### 2.1. Cultivation and Inoculation

Spinach was cultivated in a field provided by the Department of Food Science and Technology, the Faculty of Food Science and Nutrition, Bahauddin Zakariya University, Multan, Pakistan, and harvested after visual assessment, wherein leaves at various growth stages were categorized as 1st stage (i.e., baby leaves), 2nd stage (i.e., coarse stage) and 3rd stage (i.e., maturation). Leaves were cut with a sickle, washed with running tap water, dried, shredded and stored in a cool and dark place. After that, the spinach stock was divided into two different lots: one lot of leaves was placed in a cabinet dryer at 50 °C for dehydration; and the 2nd lot was separated for fermentation. *Lp. plantarum* ATCC 8014 procured from Fengchen Group Co., China, was used as the fermentation organism. A microbial culture was activated in MRS broth to produce the stock culture at 37 °C for 48 h. Inoculum was prepared in 5 mL MRS broth inoculated with 100 µL of the stock culture. Post the 48 h incubation, freshly produced bacterial cells were harvested and separated by centrifugation at 6000× *g* for 10 min. Bacterial cells recovered as sediments in falcon tubes were washed twice by suspending them in sterile normal saline. This washing solution was separated again by centrifugation, and the bacterial cells recovered were re-suspended in sterile distilled water to obtain a final cell count of 10^−6^ CFU per mL. The spinach lot separated for fermentation was immersed in sterile distilled water inoculated with a freshly prepared *Lp. planatrum* culture at the rate of 10^−6^ CFU per g. The jar was kept airtight and incubated at 37 °C for 120 h. Fermented spinach leaves were harvested from the container, spread on nylon-mesh trays and dried in a cabinet dryer at 45 °C. The dried leaves were ground with heavy-duty mills. Microbial viable counts were checked using the plate count method on DeMan, Rogosa, Sharpe (MRS) agar plates (oxoid, UK). Agar plates were incubated for 24 h intervals at 37 °C. The colonies that appeared were counted using a colony counter and the results were expressed as CFU per g. Each sample was replicated three times and mean counts were presented.

### 2.2. Determination of Acidity, pH and Sugars

Ten grams of spinach powder was added to 100 mL distilled water to obtain a homogenized mixture, and pH was evaluated using the pH meter (TS-PH200, portable). Titratable acidity and pH of the samples were evaluated following the official methods (947.05, 973.41) as prescribed in AOAC, 2012 [19].

### 2.3. Determination of Nutritional Composition

AOAC, 1990 [20] official methods were adopted to determine the proximate composition of samples. The moisture content of the spinach was evaluated using the hot-air oven method until a constant weight was obtained, using method no. 925.09. Fat content was evaluated using the Soxhlet apparatus, where *n*-hexane was used as a solvent following method no. 920.39. Protein content was determined using the micro-Kjeldahl procedure with acid digestion of the sample and subsequent alkaline distillation using a conversion factor of 6.25, following method no. 979.09. Ash content was evaluated via heating at 550 °C in a muffle furnace, until grayish-white ash was obtained, using method no. 923.03. Fiber was calculated via acid-base digestion with 1.25% H_2_SO_4_ and 1.25% NaOH, using method no. 985.29.

### 2.4. Total Phenolic Content and Antioxidant Potential

Fermented and non-fermented spinach powder was separately added to test tubes and each sample was homogenized in *n*-hexane, ethanol and water and centrifuged for 20 min at 140× *g* at 4 °C. This procedure was repeated three times and supernatants of both samples, i.e., fermented and non-fermented, were collected in separate tubes.

#### 2.4.1. Total Phenolic Contents

Total phenolic contents (TPC) were evaluated using the procedure adopted by Qamar et al. [21]. Methanol spinach extract (100 μL) was diluted with distilled water (3 mL) and by adding Folin–Ciocalteau reagent (0.5 mL). Subsequently, 20% Na_2_CO_3_ was added, mixed properly and kept for 30 min. Sample absorbance was measured at a wavelength of 765 nm using a spectrophotometer (V-3000; VWR, Darmstadt, Germany) and gallic acid was used as a standard. TPC was calculated as mg gallic acid equivalent (GAE)/100 g dry-weight (DW) using ethanol as blank.

#### 2.4.2. DPPH Assay

Free radical-scavenging capability was assessed using the DPPH (2,2-diphenyl-1-picrylhydrazyl) radical [22]. Methanolic extract of spinach was added to 3.9 mL of DPPH (25 mg/L) in methanol. Absorbance of the blank sample was measured at 515 nm using methanol without DPPH. Sample absorbance was calculated in μmoL trolox equivalent (TE)/g until reaction touched the plateau. The following Equation (1) was used to calculate the % inhibition of DPPH:Inhabitation% = (A_0_ − A)/A_0_(1)
where,

A_0_ = Beginning absorbance of sample at 515 nm

A = Sample final absorbance at 515 nm

Final outcomes are given in μmol TE/g as DW.

#### 2.4.3. FRAP Assay

Ferric-reducing antioxidant power (FRAP) analysis was determined by adopting the method of Qamar et al. [23]. FRAP reagent contained 300 mmol/L acetate buffer (pH 3.6), 10 mmol/L 2,4,6-tripyridyl-s-triazine (TPTZ) in 40 mmol/L HCl, and 20 mmol/L FeCl_3_, in a ratio of 10:1:1. FRAP reagent (3 mL) was drawn in the test tubes, further adding 100 μL of extract sample, and vortexed. Sample absorbance was measured at 593 nm after 4 min. The results were expressed in μmol of TE per gram DW (μmol TE/g).

### 2.5. Antinutrient Factors

#### 2.5.1. Oxalates

The oxalate content of the spinach sample was assessed, adopting the method followed by Amin et al. [24]. One gram of the methanol sample was drawn in a 100 mL conical flask, 75 mL 3N H_2_SO_4_ were added, continuously stirred with a magnetic stirrer for 1 h, and filtered using Whatman No. 1 filter paper. Twenty-five milliliter filtrate was taken in a 100 mL flask and titrated while heating at 80–90 °C with 0.1 N KMnO_4_ solution until a pink color was steady for at least 30 s. This procedure was replicated for every sample in triplicate and notes were made of the results [24].

#### 2.5.2. Phytates

Phytate contents in spinach powder samples were determined in accordance with Haug and Lantzsch [25]. Two grams of finely ground sample was immersed in 20 mL of HCl (0.2 N) and filtered. About 0.5 mL filtrate was drawn in the test tube, subsequently added with 1 mL of NH_4_Fe(SO_4_)_2_ solution, the solution was boiled in a water bath for 30 min, abruptly cooled down with ice for 15 min, centrifuged at 1260× *g* for 15 min, and the supernatant gathered. A test tube was filled with 1 mL of supernatant, 1.5 mL of pyridine solution was added and the absorbance of the sample at 519 nm was measured. A standard-curve extrapolation was used to determine the sample’s phytate content using phytic acid standard solutions.

#### 2.5.3. Alkaloids

Alkaloids content of the sample were determined in accordance with Onwuka [26]. Five grams of the sample was added to 50 mL of acetic acid (10%) solution in ethanol. The mixture was shaken properly, left for 4 h at room temperature, filtered and the filtrate evaporated until it was one-fourth of its actual volume. Concentrated NH_4_OH was drawn gradually (dropwise) to attain alkaloids precipitate. These precipitates were filtered with weigh filter paper and washed with NH_4_OH (1%) solution.

#### 2.5.4. Saponins

The saponin contents of samples were calculated following the procedure described by Obadoni and Ochuko [27]. About 250 μL of the sample was drawn in the test tube, the reagent mixture (sulfuric acid/glacial acetic acid, 1:1 *v*/*v*) was added, vigorous shaking was performed for 30 s until color development occurred, and it was heated for 30 min at 60 °C. During the heating reaction, a purple color developed; the sample was subsequently cooled in chilled water. The absorbance of the sample was measured at 527 nm. Saponin contents (mg/mL) were calculated by comparison with the oleanolic acid standard curve (range of 100–1000 μg/mL).

### 2.6. Determination of Nitrates and Nitrites

Nitrate and nitrite levels in non-fermented and fermented spinach samples were determined, adopting the method followed by Merino [28]. One milligram of the spinach powder was placed in an Erlenmeyer flask holding 60 mL hot water (50–60 °C). The sample was mixed adequately to generate a homogenous suspension. Subsequently, 4 mL of zinc acetate dihydrate (230 g in 1000 mL) and 4 mL of potassium hexacyanoferrate (II) trihydrate {150 g K_4_[Fe(CN)_6_]·3 H_2_O mixed with water and diluted up to 1000 mL} were added to the flask and mixed. Homogenates were transferred to falcon tubes and centrifuged for 10 min at 4000 rpm. Filtrate were pipetted as clear supernatant and diluted with 100 mL water in a volumetric flask.

For the determination of nitrate levels, approximately 20 mL of the above diluted sample was drawn in a 100 mL volumetric flask and further mixed with 10 mL of ammonia buffer with continuous stirring. Afterwards, 2 mL of the aqueous acidic (HCl) sulphanilamide (dissolved 2 g sulphanilamide in distilled water and 105 mL HCl) was added and the mixture was homogenized and incubate for 5 min at room temperature. Approximately 2 mL of the diluted *N*-(1-naphthyl)-ethylenediamine dihydrochloride solution was added to the mixture. The absorbance at 540 nm was used in the Equation (2):(2)$$NO-2=\frac{Abs(s)-Abs(bl)}{b1}\times F$$
where,

*NO*_2_: Nitrogen dioxide

*Abs*(*s*): Sample absorbance

*Abs*(*bl*): Blank absorbance

*b*1: Calibration graph

*F*: Dilution factor

Regarding the reduction of nitrate to nitrite, twenty milliliter of the spinach sample extract (obtained from nitrate protocol) was taken into a glass bottle and added with 10 mL ammonia (25%) and 0.1 g of zinc sulphate powder. The contents of the bottle were shaken vigorously for 5 min, manually or with the use of a shaker. The clear supernatant obtained on standing the suspension was filtered through the filter paper. The resulting filtrate was collected in a 100 mL volumetric flask. Approximately 2 mL of the aqueous acidic (HCl) sulphanilamide coloring agent was added into the mixture and mixed thoroughly. Subsequently, 2 mL of the diluted *N*-(1-naphthyl)-ethylenediamine dihydrochloride solution was added and the mixture diluted to 200 mL. The absorbance of each sample was measured at 540 nm, according to Equation (3).
(3)$$NO-3=\frac{Abs(s2)-Abs(bl2)}{b2}\times F$$
where,

*Abs*(*s*): Sample absorbance

*Abs*(*b*2): Blank absorbance

*bl*2: Calibration graph

*F:* Dilution factor.

### 2.7. Determination of Glucosinolates

Glucosinolates content in spinach samples were determined according to Mawlong et al. [29]. An 0.2 g sample of defatted spinach was placed in a 2 mL vial containing 80% methanol. Subsequently, overnight incubation was performed, followed by sample drying. Once again, the sample was mixed with 80% methanol, allowed to stand overnight and centrifuged for 4 min at 3000 rpm at 37 °C. Following centrifugation, the supernatant was collected and made up to a total volume of 2 mL with 80% methanol. For estimation, 100 µL of extract was mixed with 0.3 mL of double-distilled water and 3 mL of 2 mM sodium tetrachloropalladate (58.8 mg sodium tetrachloropalladate, 170 µL concentrated HCl and 100 mL double-distilled water). Following 1 h of incubation at room temperature, the absorbance was measured at 425 nm with a spectrophotometer (V-3000; VWR, Darmstadt, Germany). The same process was used for blank (i.e., without the extract). Total glucosinolates were estimated by putting each sample’s optical density (OD) at 425 nm into the following Equation (4):y  =  1.40  +  118.86  ×  OD_425_(4)

### 2.8. Statistical Analysis

The results were presented as mean ± standard deviation (SD). The obtained data were statistically analyzed using completely randomized design (CRD). Biological activities were assessed by employing analysis of variance (ANOVA). Means were compared by least significant difference at *p* < 0.05 as level of significance (Statistix 8.1, Tallahassee, FL, USA) version was used to perform the data analysis.

## 3. Results

As expected, the pH value of spinach leaves was significantly (*p* < 0.05) reduced from 5.0 ± 0.75 to 3.4 ± 0.05 at 0–48 h of fermentation time. Similarly, the data on pH values at the emerging growth stage showed the decline of pH values from 3.6 to 3.4 as the fermentation time was increased from 72 to 120 h. However, in contrast to the pH mean values, the acidity mean values were significantly (*p* < 0.05) increased as the fermentation time was increased from 0 to 120 h, *viz*. 110, 230, 380, 760, 830 and 900 mg/100 g at the emerging growth stage, whereas the values were 120, 240, 390, 780, 840 and 910 mg/100 g at the mature growth stage, and 110, 230, 380, 760 and 830 and 900 mg/100 g at the coarse growth stage of spinach samples. The results for the effect of fermentation time on the carbohydrate concentrations of spinach samples revealed, as expected, a significant (*p* < 0.05) decrease in content from 46 ± 6.18 to 35.4 ± 5.53 g/100 gm. The increase in total viable counts was unfolded as increasing the fermentation time, which further declined after 72 h at the 1st, 2nd and 3rd growth stages. These results are illustrated in Table 1.

The data given in Table 2 indicates a significant difference in approximate composition of the spinach samples harvested at different growth stages, alongside the significant effect of fermentation on different nutritional indices of the spinach. The results showed the significant (*p* < 0.05) effect of fermentation on increasing the protein and fiber content of spinach from 2.53 to 3.53% and 19.33 to 22.03%, respectively, and on reducing total carbohydrate content from 52.92 to 40.52%. Such effects were consistent in all three growth stages. Estimating the moisture content of the spinach is considered critical for the long-term storage of the powder. Antinutrients reduce the availability of nutrients directly or indirectly through metabolic pathways. These antinutrients’ (known as “Allelochemicals”) distribution and quantity varies with plant species and the variety and type of vegetable, among other factors. Common antinutrients in leafy greens include saponins, nitrates, nitrites, oxalates, phytates, tannins, alkaloids and glucosinolates. In non-fermented spinach powder, saponin content slightly declined from 10.59 ± 0.96 (1st stage) to 10.56 ± 0.48 mg/100 g (3rd stage). Meanwhile, saponin content in fermented spinach powder slightly declined from 10.55 ± 0.04 to 10.37 ± 0.41 mg/100 g after fermentation with *Lp. plantarum*. Alkaloid content was reduced significantly as a result of fermentation as illustrated in Table 3.

A minimum level of alkaloids was observed in the fermented second stage (2.33 ± 0.15 µmol/g) when compared to the non-fermented second stage where alkaloid content was 3.17 ± 0.01 µmol/g. Glucosinolates content reduced significantly as a result of fermentation: 93.0 to 3.50 µmol/g (1st stage), 101.5 to 6.50 µmol/g (2nd stage) and 201.2 to 10.50 µmol/g (3rd stage). Nitrite and nitrate content reduced significantly as result of fermentation from 2.61 to 1.36 mg/100 g and 0.951 to 0.220 mg/100 g, respectively, at the 3rd stage.

Fermentation resulted in a significant decline in oxalate content (Table 3). In the 1st stage, oxalate content (0.07 ± 0.06 mg/100 g) was significantly (*p* < 0.05) lower than in the 3rd stage (0.04 ± 0.06 g/100 g) of the non-fermented spinach sample. Importantly, when the 3rd stages of non-fermented and fermented were compared, a significant decline was observed from 0.07 ± 0.06 mg/100 g to 0.02 ± 0.011 mg/100 g, respectively.

The total phenolic content of fermented spinach at 1st, 2nd and 3rd growth stages were 31.6, 30.5 and 29.6 mg/100 g DW, respectively. However, the concentrations of non-fermented spinach were higher in spinach samples taken at the 1st growth stage (i.e., 24.9 mg GAE/100 g), while the minor mean values of phenolics were measured at the 3rd growth stage (i.e., 23.7 mg GAE/100 g). The overall group means of the solutions showed that the ethanol extract has the highest number of total phenolics (i.e., 36.4 mg GAE/100 g), followed by *n*-hexane (i.e., 27.4 mg GAE/100 g) and water (i.e., 21.3 mg GAE/100 g).

DPPH scavenging activity of the fermented and non-fermented spinach samples are depicted in Table 4. The ability to scavenge stable free radicals was the highest at the 3rd stage (76.60% inhibition) of the *n*-hexane-fermented spinach extract followed by the 2nd stage (76.02% inhibition) and 1st stage (73.7% inhibition). On the other hand, the ethanol-non-fermented spinach extract at the 3rd stage displayed more scavenging capacity (82.5% inhibition), followed by the 1st stage (76.6% inhibition) and 2nd stage (75.8% inhibition). 

The results showed that the highest FRAP values were recorded in the ethanol-non-fermented spinach extract at the 3rd stage (42.9 μmoL TE/100 g), followed by the 2nd growth stage (42.3 μmoL TE/100 g) and 1st growth stage (40.8 μmoL TE/100 g). On the other hand, results of reducing potential were a little lower in fermented samples. Non-fermented spinach extract at all three stages showed excellent reducing properties (37.4–37.9 μmoL TE/100 g), but results were not significant between the growth stages. On the other hand, water-fermented spinach extract outlined comparable reducing potential to non-fermented ethanol spinach extract in all three growth stages (48.8 μmoL TE/100 g). It can be seen that the antioxidant potential of both fermented and non-fermented samples are almost the same and fermentation exerted no negative impact.

## 4. Discussion

Lactic acid bacteria are naturally present in most of vegetables [30]. The results showed that after fermentation the viable counts reached a general value of 7–8 log cycle [log(CFU/g)] (*p* > 0.05), proving the good performance of lactic acid fermentation in spinach (Table 1). During fermentation, carbohydrate content is metabolized through lactic acid fermentation. Homofermentative LAB metabolizes carbohydrates into lactic acid, whereas heterofermentative LAB metabolizes carbohydrates into lactic acid, acetic acids (or ethanol) and carbon dioxide. In this study, *Lp. plantarum*, a facultative heterofermenter, gradually decreased the pH value, creating the acidic environment which inhibits the growth of many other microorganisms including foodborne pathogens [31,32,33]. Fermentation processes not only produce organic acids but also some other substances like reuterin, acetoine, acetaldehyde, ethanol, diacetyl, and bacteriocins. They act as biopreservative agents which hinder the growth of spoilage, and non-pathogenic and pathogenic microorganism [34]. These substances enhance the flavor, aroma and texture of food. Hence, fermented vegetables are significantly safer for consumption and have a longer shelf-life when compared to non-fermented products. The type of the bacterial fermentation depends upon the nature and type of the vegetables [35]. In addition to the biologically active compounds released during fermentation, the lowered pH value promotes the growth of fermentation bacteria in the range of 5.0–3.4, inhibiting the multiplication of harmful microbes in food. The rate and degree of progressive acidity of plant material usually depends on the susceptibility and nature of the fermented product. Dallal et al. [36] reported that traditional Iranian fermented vegetables have a pH value of 4.0, whereas pH of 4.4 to 4.7 was evaluated in kimchi by Choi et al. [37]. In the current investigation, fermented spinach has pH values between 5.0 to 3.4 at all three growth stages. These low pH values are a good indicator of the development of lactic acid fermentation.

Non-fermented spinach powder had high values of carbohydrate content when compared to fermented spinach powder (Table 2). Carbohydrate content was reduced after fermentation due to its consumption as a nutrient by the microorganism responsible for metabolic and growth activities [38]. Fermentation initiates the hydrolyzing enzymes of starch like maltase and α-amylase which break down the starch into simple sugar and maltodextrin. It was also reported that total carbohydrate content of the fermented spinach powder reduced because the released glucose in fermentation processes is further used during the microbial fermentation [39].

Moisture content in samples depends on the nature and water-holding capacities of the powder. High moisture content lessens the shelf-life since it enhances the susceptibility to microbial spoilage [40]. In the present investigation, it can be seen that fermented and non-fermented samples have no significant difference in moisture content.

Fermented spinach powder showed lower ash content when compared to non-fermented groups of spinach at different growth stages. A high amount of ash content in non-fermented spinach powder implies that they have a rich source of inorganic elements [41]. In the current research, it was illustrated that fermentation has a negative impact on the ash content and lowers the mineral element in the sample. The reduction in the ash content of fermented samples is due to the leaching down of soluble mineral elements during fermentation [42]. Higher or lower amounts of ash content in samples depend upon the fermentation type, processing conditions and nature or type of vegetable. Previously, the ash content decreased from 29.62% to 24.13% because the submerged fermentation technique that was employed leached down the mineral elements in water [43]. The current investigation identified an increased fat content in fermented spinach samples. A similar impact of fermentation was reported by Ifesan et al. [44] when investigating the pseudocereal amaranth, where fat content increased from 20 to 40%. The increase in fat can be associated with the fermentation method, which elevates fat content due to the breakdown of bonds between triglycerides and other acylglycerols, and to the conversion of larger fatty acids into smaller ones [45].

The protein content of fermented spinach samples increased when compared to non-fermented spinach samples because of fermentation hydrolysis and the release of embryonic proteins required for seed germination [39]. The current investigation results are in line with the study by Afoakwa et al. [46], who reported an increase in protein content due to fermentation. Nutritionally, a higher level of protein in foods is favorable for the development and growth of the children. Protein is a vital element for organ- and body-tissue-building and biochemical activities [47]. Plant protein digestibility increases through fermentation [39].

The higher fiber content in samples is linked with the upsurge in moisture-retention [36]. The fiber content in samples ranges from 14.43 ± 0.10 to 22.02 ± 0.07%, and the maximum fiber content was found in the fermented samples (16.55 ± 0.37 to 22.02 ± 0.07%). Carbohydrates are the main source of energy in the human body. Non-fermented spinach samples had a higher carbohydrate content when compared to the fermented spinach samples. Reduction in carbohydrate content is due to utilization of starch during the metabolic activities of the fermentative microorganism and results in lower concentrations [39].

Saponins are heat-sensitive compounds that are also soluble in water, which are lowered after soaking and blanching [48]. Previous investigations described how saponins perform inhibitory activities against digestive enzymes like lipase, trypsin, amylase and glucosidase, creating health disorders associated with indigestion [49,50]. Saponins also reduce mineral- and vitamin-absorption owing to their ability to form complex structures like fat-soluble vitamins. In contrast, saponin content was also reported to perform biological activities which are structure-dependent [51]. Reduction in saponin content after fermentation may be attributed to the action of β-Glucosidase which catalyzes the structural degradation of saponins, resulting in their removal from the plant matrix [52]. In the current investigation, a slight reduction was observed in the saponin content (Table 3)

Glucosinolates are naturally occurring components that have a pungent taste (due to their sulfur-containing elements) and are present in various plants and vegetables like mustard, spinach and cabbage [53]. The hydrolyzation products of glucosinolates have great interest because they possess a positive impact on human wellbeing. They are also supportive in improving the capability of the liver to neutralize toxic elements and, subsequently, the prevention of ovarian and breast cancer [54,55]. Fermentation and drying techniques showed that fermentation was most appropriate for the reduction of glucosinolates (Table 3). Similarly, when spinach was separated for fermentation, significant variation was observed in the glucosinolate content: it decreased from 3.50–10.5 µmol/g when compared to the fresh sample (recorded with 3–201 µmol/g). Previously, fermentation led to the reduction of glucosinolates in *Brassica* due to the enzymatic conversion of glucosinolates into sulfur and glucose [55].

Oxalates are natural chemical compounds which are present in plants. They can form when oxalic acid interacts with potassium, sodium, calcium and magnesium. Oxalic acid is generated during processing or digestion, and it binds to nutrients, making them unavailable to the body. Oxalates hinder the bioavailability of nutritional components, decreasing the nutritional value of food. Some fruit and vegetables contain high amounts of oxalates like cauliflower, parsley, radish, spinach, beets, blueberries beans and blackberries [56]. Oxalates have a negative effect on the bioavailability of calcium and magnesium. In the current investigation, oxalate content was reduced due to *Lp planetarium fermentation* from 0.07 ± 0.006 to 0.02 ± 0.011 g/100 g (Table 3). Oxalates are water-soluble components and they leach down via blanching, boiling and steaming [57]. Previously, Hassan et al. 2015 [58] observed a similar decline in oxalate content in leafy vegetables, ascribed to the hydrolytic action of enzymes which occurred in fermentation [58]. Hence, it was noted that fermentation is helpful in reducing oxalate content in spinach. Alkaloids are responsible for the sharp taste in leafy greens.

Phytate is the primary phosphorus storage form in plants. Phytates have been found to alter nutritional availability in plant diets by creating mainly stable complexes with macronutrients like carbohydrates and protein, which render them available for digestion and absorption. Research conducted by Lee et al. 1993 [59] was performed on rats and the results showed that the metabolism of zinc, calcium and phosphorus was highly affected by phytates [58]. Different studies have proved that phytate hinders the absorption of zinc, thus resulting in deficiency of zinc (Table 3).

Phytate-reduction increases the availability of soluble minerals by multiple orders of magnitude and also increases the activity of various phenolic compounds. When spinach was fermented, the phytate concentration decreased significantly, with an even greater decline as the fermentation duration progressed [60]. Nitrates and nitrites are considered antinutrient components of the food, exhibiting deleterious impacts on humans if utilized over the permissible limit. A higher level of nitrites was observed in non-fermented spinach powder (5.81 ± 0.010 µmol/g DW) at the 3rd stage, while lower values were attained in the fermented 3rd stage (2.61 ± 0.001 µmol/g DW). The outcome of the current investigation slightly resembles the previous research conducted by Wang et al. [61].

The antioxidant potential of spinach and its therapeutic role against various health disorders (such as stomach problems, colon, prostrate and liver cancers, cardiovascular disorders and digestive problems) can be better understood by exhibiting its inhibitory impact and protective role. Phenolic compounds are natural phytonutrients known for redox properties which help these natural compounds to act as antioxidants [62,63,64,65,66]. The spinach exhibited the lowest content of total phenolics in both fermented and non-fermented groups prepared with water when compared with all other solutions (Table 4). An earlier study by Turkmen et al. [67] reported roughly comparable findings for phenolics which ranged from 183 to 1344 mg GAE/100 g in fresh green leafy vegetables.

## 5. Conclusions

Spinach has outstanding nutritional value and health-promoting biological characteristics. It is one of the most highly valued leafy greens and a top food option for consumers globally. Fermentation of vegetables through lactic acid bacteria is a biopreservative approach that produces high-quality (and safer) food. In our investigation, this was revealed by contrasting nutritional and oxalate levels with increasing plant age. The fermentation of spinach at different growth stages with *Lp. plantarum* enhanced the nutritional composition and nutrient bioavailability of the elements by lowering the antinutritional factors. Fermentation is a cost-effective microbial approach to minimizing the burden of intrinsic toxicants in spinach, improving nutritional bioavailability and increasing nutrient-delivering properties. Fermentation degraded the detrimental components from the spinach samples, i.e., reduced the antinutrients and increased the absorption of the metabolites. These findings strongly support that the application of lactic acid fermentation in spinach at different growth stages is a safer alternative to conventional processes that are commonly used, chiefly chemical preservation techniques and invasive approaches that might compromise the organoleptic, nutritional and safety aspects of consumer products.

## Figures and Tables

**Table 1 microorganisms-11-02343-t001:** Effect of fermentation on pH, acidity, carbohydrate content and total viable count at various growth stages (1st, 2nd, 3rd) of spinach.

Stage	Fermentation Time(h)	pH	Acidity (mg/100 g LA)	Carbohydrates (mg/100 g)	Total Viable Counts [Colony-Forming Units (CFU)/g]
**1st**	**0**	5.0 ± 0.75 ^a^	110 ± 0.01 ^e^	46.0 ± 6.18 ^a^	8 × 10^−5^ ± 1.25 ^d^
	**24**	4.8 ± 0.74 ^a^	230 ± 0.02 ^e^	42.0 ± 4.46 ^ab^	7.8 × 10^−6^ ± 1.22 ^d^
	**48**	3.7 ± 0.58 ^b^	380 ± 0.03 ^cd^	41.6 ± 4.16 ^ab^	7.6 × 10^−6^ ± 1.19 ^d^
	**72**	3.6 ± 0.56 ^b^	760 ± 0.08 ^b^	39.1 ± 3.66 ^ab^	1.3 × 10^−8^ ± 1.89 ^bc^
	**96**	3.4 ± 0.53 ^b^	830 ± 0.11 ^ab^	38.0 ± 7.24 ^ab^	1.2 × 10^−7^ ± 1.60 ^abc^
	**120**	3.4 ± 0.05 ^b^	900 ± 0.15 ^ab^	36.4 ± 4.21 ^ab^	1.1 × 10^−7^ ± 1.74 ^a^
**2nd**	**0**	5.1 ± 0.12 ^a^	120 ± 0.02 ^e^	46.0 ± 7.90 ^a^	8 × 10^−4^ ± 0.75 ^d^
	**24**	4.9 ± 0.39 ^a^	240 ± 0.07 ^de^	43.0 ± 6.45 ^ab^	7.7 × 10^−5^ ± 1.47 ^d^
	**48**	3.8 ± 0.18 ^b^	390 ± 0.03 ^c^	42.6 ± 6.56 ^ab^	7.5 × 10^−6^ ± 0.86 ^d^
	**72**	3.7 ± 0.20 ^b^	780 ± 0.06 ^ab^	39.0 ± 6.09 ^ab^	14 × 10^−7^ ± 4.09 ^c^
	**96**	3.5 ± 0.55 ^b^	840 ± 0.07 ^ab^	37.0 ± 5.78 ^b^	1.3 × 10^−7^ ± 1.81 ^abc^
	**120**	3.5 ± 0.55 ^b^	910 ± 0.18 ^a^	35.4 ± 5.53 ^ab^	10 × 10^−6^ ± 2.18 ^ab^
**3rd**	**0**	5.0 ± 0.78 ^a^	110 ± 0.01 ^e^	46.0 ± 0.72 ^a^	8.2 × 10^−5^ ± 1.36 ^d^
	**24**	4.8 ± 0.22 ^a^	230 ± 0.02 ^e^	42.0 ± 0.99 ^ab^	8 × 10^−6^ ± 1.37 ^d^
	**48**	3.7 ± 0.17 ^b^	380 ± 0.03 ^cd^	41.6 ± 3.31 ^ab^	7.6 × 10^−6^ ± 2.20 ^d^
	**72**	3.6 ± 0.37 ^b^	760 ± 0.08 ^b^	39.1 ± 1.89 ^ab^	1.3 × 10^−7^ ± 1.28 ^bc^
	**96**	3.4 ± 0.36 ^b^	830 ± 0.03 ^ab^	38.0 ± 2.08 ^ab^	1.1 × 10^−7^ ± 1.31 ^abc^
	**120**	3.4 ± 0.34 ^b^	900 ± 0.18 ^ab^	36.4 ± 5.68 ^b^	1.0 × 10^−7^ ± 1.57 ^a^

Values are given as mean ± standard deviation (*n* = 3). LA: lactic acid. The values having similar letters in a column are not statistically significant at *p* > 0.05.

**Table 2 microorganisms-11-02343-t002:** Chemical analysis of fermented and non-fermented samples.

Groups	Stages	Moisture (g/100 g)	Ash (g/100 g)	Carbohydrates (g/100 g)	Fat (g/100 g)	Fiber (g/100 g)	Protein (g/100 g)
**Fermented**	**1st**	3.10 ± 0.10 ^d^	24.83 ± 0.21 ^d^	40.50 ± 3.40 ^b^	2.99 ± 0.10 ^a^	16.55 ± 0.37 ^d^	2.60 ± 0.10 ^b^
	**2nd**	4.20 ± 0.10 ^c^	24.13 ± 0.21 ^e^	40.52 ± 4.16 ^b^	3.01 ± 0.19 ^b^	17.37 ± 0.47 ^c^	2.80 ± 0.10 ^b^
	**3rd**	5.90 ± 0.10 ^a^	26.07 ± 0.25 ^c^	40.52 ± 3.40 ^b^	3.50 ± 0.01 ^b^	22.02 ± 0.07 ^a^	3.23 ± 0.20 ^a^
**Non-Fermented**	**1st**	3.00 ± 0.10 ^d^	27.22 ± 0.48 ^b^	49.29 ± 0.19 ^a^	1.20 ± 0.02 ^e^	14.43 ± 0.10 ^f^	1.20 ± 0.10 ^d^
	**2nd**	4.18 ± 0.02 ^c^	29.73 ± 0.04 ^a^	50.07 ± 0.40 ^a^	1.82 ± 0.02 ^c^	15.07 ± 0.10 ^e^	1.73 ± 0.15 ^c^
	**3rd**	5.66 ± 0.06 ^b^	29.62 ± 0.40 ^a^	52.92 ± 0.34 ^a^	2.59 ± 0.02 ^c^	19.33 ± 0.10 ^b^	2.53 ± 0.25 ^b^

Values are given as mean ± standard deviation (*n* = 3). The values that have similar lettering in a column are not statistically significant at *p* > 0.05.

**Table 3 microorganisms-11-02343-t003:** Antinutritional factors in non-fermented and fermented spinach from different stages (dry-weight basis).

Groups	Stages	Saponins (mg/100 g)	Glucosinolates (µmol/g)	Alkaloids (mg/100 g)	Oxalates (mg/100 g)	Phytate (mg/100 g)	Nitrite (mg/100 g)	Nitrate (mg/100 g)
**Fermented**	**1st**	10.37 ± 0.412 ^a^	3.50 ± 0.50 ^e^	3.16 ± 0.23 ^b^	0.02 ± 0.006 ^b^	0.57 ± 0.08 ^a^	1.36 ± 0.002 ^c^	0.220 ± 0.007 ^b^
	**2nd**	10.33 ± 0.451 ^a^	6.50 ± 0.501 ^d^	2.33 ± 0.151 ^d^	0.03 ± 0.006 ^a^	0.25 ± 0.011 ^a^	1.25 ± 0.003 ^b^	0.549 ± 0.004 ^c^
	**3rd**	10.55 ± 0.042 ^a^	10.50 ± 0.502 ^c^	2.20 ± 0.180 ^b^	0.02 ± 0.011 ^a^	0.43 ± 0.032 ^a^	2.61 ± 0.001 ^d^	0.951 ± 0.008 ^b^
**Non-Fermented**	**1st**	10.59 ± 0.961 ^a^	93.0 ± 2.841 ^b^	7.67 ± 0.311 ^a^	0.04 ± 0.006 ^b^	0.74 ± 0.080 ^a^	3.56 ± 0.002 ^cd^	2.37 ± 0.001 ^d^
	**2nd**	10.57 ± 0.610 ^a^	101.5 ± 3.043 ^c^	4.35 ± 0.011 ^c^	0.07 ± 0.001 ^c^	0.85 ± 0.021 ^a^	4.66 ± 0.002 ^d^	4.67 ± 0.009 ^b^
	**3rd**	10.56 ± 0.481 ^a^	201.2 ± 2.471 ^a^	6.45 ± 0.020 ^d^	0.07 ± 0.006 ^a^	1.97 ± 0.081 ^a^	5.81 ± 0.010 ^a^	5.27 ± 0.001

Values are given as mean ± standard deviation (*n* = 3). The values that have similar lettering in a column are not statistically significant at *p* > 0.05.

**Table 4 microorganisms-11-02343-t004:** Antioxidant activity of non-fermented and fermented spinach at different growth stages.

Group	Stage	(DPPH) (% Inhibition)			(FRAP) (µmol TE/100 g)				(TPC) (%)	
		Ethanol	*n*-Hexane	Water	Ethanol	*n*-Hexane	Water	Ethanol	*n*-Hexane	Water
**Fermented**	1st	66.49 ± 0.01 ^j^	73.74 ± 0.05 ^e^	72.21 ± 0.02 ^g^	31.34 ± 12.03 ^ab^	12.12 ± 21.41 ^bc^	42.9 ± 1.87 ^a^	35.21 ± 7.11 ^abcd^	24.97 ± 11.50 ^cde^	26.3 ± 7.21 ^bcde^
	2nd	69.20 ± 0.03 ^h^	76.02 ± 0.03 ^b^	73.26 ± 0.01 ^ef^	34.19 ± 13.18 ^ab^	13.08 ± 21.44 ^bc^	42.9 ± 1.88 ^a^	35.16 ± 7.22 ^abcd^	24.19 ± 11.48 ^cde^	26.03 ± 7.60 ^bcde^
	3rd	67.57 ± 0.01 ^i^	76.60 ± 0.10 ^b^	74.88 ± 0.00 ^cd^	34.76 ± 12.28 ^ab^	14.04 ± 21.44 ^bc^	42.9 ± 1.87 ^a^	35.12 ± 7.24 ^abcd^	23.69 ± 11.76 ^de^	26.4 ± 7.31 ^bcde^
**Non-Fermented**	1st	67.09 ± 0.24 ^ij^	76.67 ± 0.00 ^b^	73.91 ± 0.15 ^de^	40.79 ± 10.43 ^a^	15.98 ± 12.95 ^c^	37.9 ± 23.33 ^ab^	37.70 ± 4.44 ^abc^	31.62 ± 13.99 ^abcd^	26.3 ± 10.22 ^e^
	2nd	72.43 ± 0.02 ^fg^	75.87 ± 1.38 ^bc^	74.17 ± 0.01 ^de^	42.29 ± 8.03 ^a^	18.37 ± 12.62 ^c^	37.6 ± 23.31 ^ab^	37.71 ± 4.47 ^abc^	30.54 ± 13.00 ^abcd^	26.4 ± 9.90 ^e^
	3rd	72.08 ± 0.01 ^g^	82.48 ± 0.04 ^a^	73.52 ± 0.01 ^e^	42.89 ± 7.08 ^a^	20.83 ± 12.49 ^c^	37.4 ± 23.13 ^ab^	37.69 ± 4.53 ^abc^	29.59 ± 13.28 ^abcde^	26.4 ± 9.91 ^e^

Values are given as mean ± standard deviation (*n* = 3). The values that have similar lettering in a column are not statistically significant at *p* > 0.05.

## Data Availability

Not applicable.

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
