# Peer review of "Effect of Growth Stages and Lactic Acid Fermentation on Anti-Nutrients and Nutritional Attributes of Spinach (Spinacia oleracea)"

_microorganisms, 2023, doi:10.3390/microorganisms11092343_

Round 1

Reviewer 1 Report

Manuscript: Effect of growth stages and lactic acid fermentation on anti-nu- 2 trients and nutritional attributes of spinach (Spinacia oleracea)

The article explores the benefits of fermented spinachs respect to fresh spinachs. In general, the article is well written but there are some important aspects that need to be clarify until be accepted. The authors should clarify how many replicates has each experiment. Regarding the microbial population involved in the fermentation, the authors should indicate the natural contamination of the spinachs to compare with the inoculated ones. And if it is possible, also indicate the grade of implantation of the LAB used. Because now it is not clear and PCA does not discriminate the microbial population, maybe being involved in the fermentation yeasts and other kind of bacteria.

Line 25: I would not focus it on just one continent since it is spread all over the world. This increase the importance of this food.

Line 31. spinach without upper case letter. Revise throught the manuscript.

Lines 73 and 74: Add the references regarding the fermented food’s characteristics.

Line 76: LAB instead of lactic acid bacteria

Line 101: Use the abbreviation Lp. plantarum, because is the second time you mention it. What concentration you inoculate using this methodology?, this is crucial to know the impact of this bacterium in the fermentation steps.

102: The verb in the sentence for acidity, pH and sugar is not present. Rewrite the sentence please.

Line 103: Plate count method in plate count agar (PCA)?

Line 104: How many replicates did you use for the experiments?

Line 107: Brand and location? This is also missing in other parts as statistical analysis.

Line 123: Express the rpm in xg.

Line 124: The combined supernatants of fermented and non-fermented spinach power is what is then measured? Wouldn't it be better to do it separately and check that the different  batches have the same phenols? Rewrite the sentence if both batches were done independently.

Line 143: TE? Trolox equivalent? Indicate the first time you mention it and not in line 150.

Line 159: How many samples were done?

Line 165: rpm to xg

Line 171: Only one sample withouth replicates?

Statistical analysis: Which tests did you use?

Line 251: Use other expression instead of in contrast, both characteristics are related and there are not opposite things.

Table 1: This table shows the results of fermentation for different times after add the Lp. plantarum. Do you have the results for total viable counts of the spinachs without the addition of the starter culture? It would indicate the contamination of the samples since in this culture medium grows all kind of microorganisms, not only LAB. For LAB implantation, it would be better to use MRS agar.

Express the counts using one number because use 13.1 is confusing. Is more clear to use 1.3 x 10^8.

Table 3: Revise the exponents, there is one missing and other letter is not in the exponent. Use the same number of decimals.

Table 4: it is not well centered. Use the same number of decimals. Try to adapt the raws to have the full number in one raw. Maybe you can disminished the letter, make the first column smaller…

Line 330: With these results is impossible to know if there are all LABs, and if this counts represent the starter culture. It is necessary to show the natural contamination of spinachs.

Line 462: Lp. plantarum

There are some little mistakes.

Reviewer 2 Report

This is a nice laboratory work analyzing the effect of growth stages and lactic acid fermentation on composition of spinach. The idea of the analysis of different growth stages of spinach is very interesting however, authors did not explain it in introduction at all, only mentioned in the discussion and did not mention about this in conclusions. It is quite important part of the title of this paper so you should definitely refer to this. Besides, the paper is quite fair written, however it needs some explanations and corrections, below you can find the details.

You should add information in abstract that the content of fiber,  protein e.c.t. you express in dry weight, as I suppose

Line 66: what do you mean by recent world?

Line 73: it looks like we have solved all health problems in this world, give specific references or rephrase this sentence

In introduction you should explain why this particular microorganism you applied

You used strange unit for acidity in table 1, you should explain what do you mean by this %, it's not obvious for all the readers. Even stranger unit you to applied in case of carbohydrates (I'm referring here to ml)

Table 2: why sometimes you use % and sometimes g/100g, I can guess why but you should explain this, unless it is a mistake

Table 3: nitrites and nitrates expressed in μmol/ g tells nothing, typical analysis of nitrates and nitrates in vegetables are expressed in milligrams, recalculate these values. It is important especially that spinach is one of the most abundant vegetable in this compound.

Besides:

I believe Latin names should be italic, correct all of which are not, e.g. line 26

There's no need to use capital letter for the word spinach

Round 2

Reviewer 1 Report

Dear authors, the manuscript has improved now after the reviewer comments. I indicate you some minor mistakes:

Line 87: Lp. plantarum withouth parenthesis

Line 106: Lp. plantarum

Line 121: deMan? Mann 

Line 124: times

Line 269: technique. Means

Table 4: The title is missing

Line 349: p<0.05

Some little mistakes detected. 

Reviewer 2 Report

Dear Authors,

You improved your paper significantly. However, you still need to be more precise about the unit of acidity. I assume it is expressed in milligrams of lactic acid; this information should be added. Something's wrong in your analysis/calculation about nitrates. Typical concentration of nitrates in spinach starts from hundreds of milligrams per kilogram of fresh leaves and usually spinach contains thousands of milligrams of nitrates. Besides, how is it possible that 0.095 and 0.053 μmol you recalculated to 0.951 and 0.22 mg, respectively?
